# Different Adsorption Behavior between Perfluorohexane Sulfonate (PFHxS) and Perfluorooctanoic Acid (PFOA) on Granular Activated Carbon in Full-Scale Drinking Water Treatment Plants

**Yong-Gyun Park [1,†], Woo Hyoung Lee [2,†] and Keugtae Kim [3,*]**

1   Pioneer Team, Research Institute, GS E&C, 33, Jong-ro, Jongro-gu, Seoul 03159, Korea; ygpark01@gsenc.com
2   Department of Civil, Environmental, and Construction Engineering, University of Central Florida, Orlando, FL 32816, USA; WooHyoung.Lee@ucf.edu
3   Department of Environmental & Energy Engineering, University of Suwon, 17 Wauan-gil, Bongdam-eup, Hwaseong 18323, Korea
*   Correspondence: kkt38@suwon.ac.kr
†   These authors contributed equally to this work.

**Abstract:** Perfluorinated compounds (PFCs) in water have detrimental effects on human health, and the removal rate of these compounds by conventional water treatment processes is low. Given that the levels of PFCs have been regulated in many regions, a granular activated carbon (GAC) adsorption process has been used in drinking water treatment plants to maintain concentrations of PFCs, perfluorohexyl sulfonate (PFHxS), and perfluorooctanoic acid (PFOA), below 70 ng/L. However, it was found that these concentrations in the final product water in local water utilities unexpectedly increased because of inappropriate operation and maintenance methods of GAC, such as its inefficient regeneration and replacement cycle. In this study, the changes in PFC concentration were monitored and analyzed in raw and final water of two large-scale water treatment plants for eight months. Additionally, the correlation of the GAC replacement cycle with the removal efficiency of PFHxS and PFOA was investigated in a total of 30 GAC basins of two drinking water treatment plants. A lab-scale experiment with a coconut-shell-based GAC column showed the possibly different mechanism of removal between PFHxS and PFOA, indicating that the sulfonate-based PFCs may be a limiting factor in GAC replacement cycle for PFCs removal.

**Keywords:** perfluorohexyl sulfonate; perfluorooctanoic acid; granular activated carbon; advanced water treatment process; drinking water

## 1. Introduction

Micropollutants in water resources are becoming a primary concern due to their detrimental effects on human health and the relatively low rate of removal by conventional water treatment processes. These pollutants are produced in manufacturing pharmaceuticals, electronics, and industrial and agricultural chemical products [1]. Nevertheless, there is still a lack of regulations and guidelines on micropollutants in many countries because of their low concentrations (e.g., pg/L to µg/L) in aquatic systems and a lack of studies and information on their properties and toxicity [2]. One of the prominent micropollutant groups in the Republic of Korea is perfluorinated compounds (PFCs); these are organofluorine substances in which hydrogen atoms on the alkyl chain are replaced by fluorine atoms [3,4]. Their carbon-fluorine bonds provide extremely high thermal and chemical stability, and they can be bioaccumulated in the environment [5].

Since the manufacture of the first PFC product by 3M in 1947, it has been widely used in industrial and commercial applications, such as cookware coatings, refrigerants, surfactants, polymers, pharmaceutical compounds, firefighting foams, paints, lubricants,

adhesives, cosmetics, paper coatings, and insecticides [4,6–9]. PFCs are largely categorized into two groups: perfluorinated sulfonates (PFSAs) and perfluorinated carboxylic acids (PFCAs). PFSAs include perfluorohexyl sulfonate (PFHxS), perfluorooctyl sulfonate (PFOS), and perfluorooctanoic acid (PFOA), which have often been detected in rivers in the Republic of Korea [9,10]. According to previous studies, PFCs exist at high concentrations in the aquatic systems of many other countries, including the USA, Germany, Italy, Japan, and China [11–15]. Additionally, they have been reported to persist through hydrolysis, photolysis, and biodegradation in the natural environment [7,16]. Therefore, the United States Environmental Protection Agency (US EPA) mandated a health advisory limit and has maintained a total of 70 ng/L of PFOS and PFOA concentrations in drinking water since 2016 [17,18]. The Korean Ministry of Environment also started monitoring PFHxS, PFOS, and PFOA in 2018. The concentrations of these micropollutants have been kept below the same limit (70 ng/L) of PFOS and PFOA concentrations in water supplied by water utilities [19].

As reported in various studies, conventional water treatment methods, such as coagulation/flocculation/sedimentation, sand filtration, and oxidation, are not suitable for removing PFCs effectively due to their physicochemical properties [3,10,20,21]. However, advanced treatment technologies, including granular activated carbon (GAC) adsorption, nanofiltration, reverse osmosis, and ion exchange, are significantly effective in managing these concentrations in the final product water of drinking water treatment plants (DWTPs) [3,22–25]. Considering the capital and operational costs of advanced methods, GAC has generally been used to remove PFCs in many DWTPs.

This study was initiated by the high PFCs concentration monitored in Nakdong River in the Republic of Korea, where intake water is supplied to DWTPs M1 and M2 in D city. The city had 1,560,000 m$^3$/d of drinking water treatment capacity, and both plants covered approximately 64% of the total water supply capacity (M1: 200,000 m$^3$/d, M2: 800,000 m$^3$/d). PFC concentrations were analyzed in the raw and final product water of both plants. Additionally, the changes in their concentrations during each treatment process were investigated, including pre-ozonation, coagulation/flocculation, filtration, post-ozonation, and GAC adsorption. Moreover, in-depth studies of GAC adsorption for PFC removal in lab and full-scale treatment facilities were conducted to evaluate the process treatment efficiency and the optimal operation methods, including the adsorption capacity and replacement cycle.

## 2. Materials and Methods

### 2.1. Sample Collection and Experimental Procedures

Sampling was conducted at DWTPs M1 and M2 located in D city, Republic of Korea. Samples were collected from the facilities of each water treatment process, such as raw water, pre-ozonation, coagulation/flocculation/sedimentation, filtration, post-ozonation, GAC adsorption, and product water. Eighty samples were collected periodically from May to December in 2018. To prevent water quality changes, the samples were collected in a 1-L brown glass bottle and stored in a refrigerator at 4 °C until analysis.

For adsorption breakthrough experiments, a coconut-shell-based GAC column was operated at room temperature (21 °C) for 6 months. The coconut-shell-based GAC was selected with the purpose of higher adsorption capacity compared to the typical coal-based GAC which has been used in M1 and M2. As presented in Figure 1, an adsorption reactor was constructed using an acrylic cylinder with a diameter of 10 cm and height of 100 cm (15.7 L total volume, 10 L effective volume), and 5 kg of adsorbents were placed into the reactor with an empty bed contact time of 15 min and a flow rate of 1.4 m$^3$/d under a downward flow condition. The reactor was backwashed once a week. After supplying filtered water from M1 to the adsorption reactor containing coconut-shell-based GAC, sampling was periodically performed to monitor the concentrations of PFHxS, PFOS, and PFOA in influent and effluent water. No apparent pH changes were observed during the experiment. Additionally, experiments on PFC removal efficiency were performed

using a test solution containing PFHxS and PFOA (50, 40, and 30 mg L$^{-1}$ for each), and the collected filtrates were analyzed. The data of at least triplicate measurements from all samples were represented as mean ± standard deviation. The statistical significance (*p* values) of the results was calculated using the unpaired Student's test.

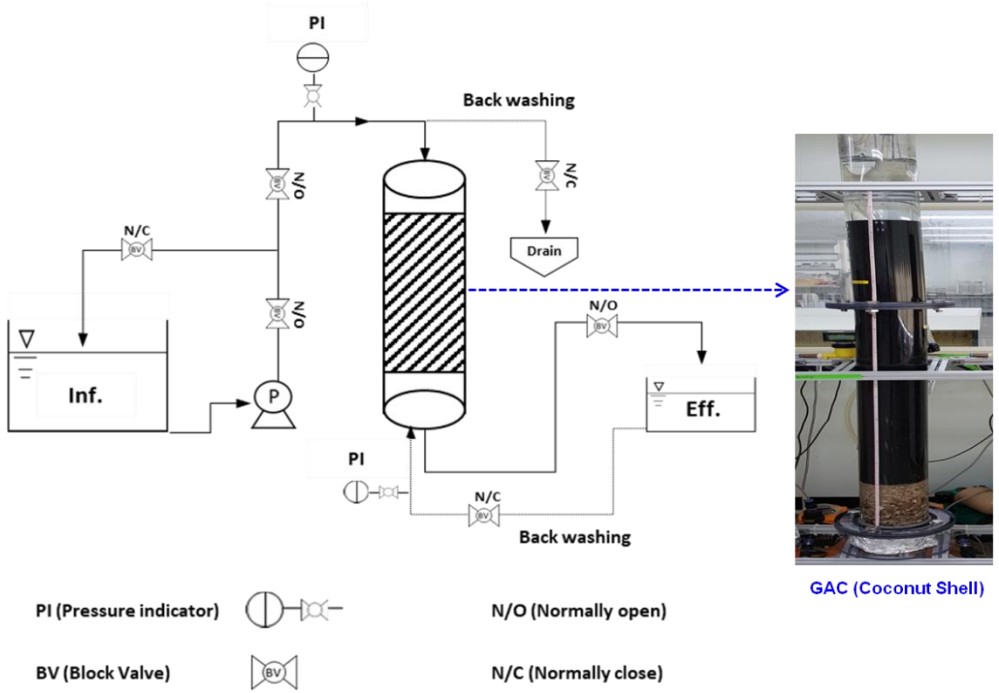

**Figure 1.** Schematic diagram of a continuous granular activated carbon (GAC) adsorption column system.

## 2.2. Materials

The PFCs used in the lab-scale experiment were PFHxS, PFOS, and PFOA, with a purity of 98% or more, which were purchased from Sigma-Aldrich (Oakville, ON, Canada). These were used to prepare stock and analytical standard solutions for the laboratory experiments. Additionally, the PFHxS, PFOS, and PFOA concentrations were quantified using standard calibration curves established with an internal standard, 13C-PFOA (98%), purchased from Perkin-Elmer Life and Analytical Science (Waltham, MA, USA). The detection limits for PFHxS, PFOS, and PFOA were 0.001 μg L$^{-1}$. High-performance liquid chromatography (HPLC)-grade ammonium acetate, methanol, acetonitrile, and Milli-Q water were purchased from Fisher Scientific (Ottawa, ON, Canada). Commercial coconut-shell-based activated carbon tested in this adsorption breakthrough experiment was purchased from Suk Korea Corporation. The physical properties of the tested coconut-shell-based activated carbon are presented in Table 1.

**Table 1.** Characteristics of GAC used for the column test.

| Substrate | Coconut Shell |
| --- | --- |
| Adsorption capacity of iodine | 1058 mL g$^{-1}$ |
| MB decolorization | 180 mL g$^{-1}$ |
| Specific surface area | 804.0 m$^2$ g$^{-1}$ |
| Micropore volume | 0.47 mL g$^{-1}$ |
| Density | 0.5 g L$^{-1}$ |
| Amount of activated carbon | 5 kg |

### 2.3. Analysis

After shaking, all samples were centrifuged at 3000 rpm for 15 min. Five milliliters of supernatant were filtered through a 0.2 μm filter (Phennomanex, Torrance, CA, USA) to remove particulate impurities prior to liquid chromatography-tandem mass spectrometry (LC-MS/MS). To avoid Teflon material containing PFCs, PP material was used in the experiments according to previous studies [14]. For the analysis of PFCs, the filtrate was directly injected into the analysis equipment using a robotic tool change model (PAS system, Zwingen, Switzerland) and ABI Sciex 5500 LC/MS/MS (AB SCIEX, MS, USA). Additional analysis conditions and PFC multiple reaction monitoring (MRM) conditions are given in Tables 2 and 3, respectively. The limits of detection and quantification in LC-MS/MS is 1.0 ng/L (n = 7, 3.143 × S.D) and 3.0~4.0 ng/L (n = 7, 10 × S.D), respectively. The extracted ion chromatogram (EIC) was used for the calibration curves of PFCs, such as PFOS, PFOA, and PFHxS, as well as for their qualitative and quantitative analyses [26].

**Table 2.** Analytical conditions of liquid chromatography-tandem mass spectrometry (LC/MS/MS) for perfluorooctanoic acid (PFOA) and perfluorohexyl sulfonate (PFHxS) analysis.

| HPLC Operating Condition | | MS/MS Operating Conditions | |
| --- | --- | --- | --- |
| **Column** | **Thermo Hypersil GOLD**<br>**50 mm × 2.1 mm × 1.9 μm** | **Ionization Mode** | **ESI [1](-)** |
| Column oven temperature | 30 °C | Curtain gas | 30 psi |
| Buffer A | 5 mM ammonium acetate water (0.02% Formic acid) | Collision gas | 12 psi |
| Buffer B | 100% methanol | Ion source gas [1] | 50 psi |
| Flow | 0.4 mL min$^{-1}$ | Ion source gas [2] | 50 psi |
| Injection | 20 μL | Ion source voltage | −4500 V |
| Run time | 10 min | Interface temp. | 600 °C |
| | | Acquisition | MRM [2] mode |

[1] Electro-spray ionization. [2] Multiple reaction monitoring.

**Table 3.** Comparison of total adsorption amount and carbon usage rate (CUR) between coal-based GAC and coconut-shell-based GAC.

| | Coal-Based GAC | | Coconut-Shell Based GAC | |
| --- | --- | --- | --- | --- |
| | **PFHxS** | **PFOA** | **PFHxS** | **PFOA** |
| Total adsorption amount (mg/ton-AC) | 276.2 | 40.2 | 1137.9 | 209.5 |
| CUR (g/d) | 0.051 | 0.050 | 0.058 | 0.037 |

## 3. Results and Discussion

### 3.1. Behavior and Fate of PFCs in DWTPs

The intake locations of both DWTPs were relatively close to each other (within 5 km, located in Nakdong River, Waegwan watershed in South Korea); thus, the concentrations of PFCs in intake water were similar. Table 4 shows the concentrations of PFHxS and PFOA in the influent and produced water from M1 and M2 for 8 months (from May to December 2018). The intake water of M1 had a PFHxS concentration of 0.023 ± 0.075 μg/L (0.000 μg/L, median) and PFOA concentration of 0.013 ± 0.008 μg/L (0.010 μg/L, median). One of the reasons for the relatively high standard deviation of the PFCs concentrations could be the seasonal changes due to heavy rain in summer (e.g., July and August). The concentrations of both compounds in the water produced from the plant were 0.052 ± 0.052 μg/L (0.028 μg/L, median) and 0.007 ± 0.009 μg/L (0.005 μg/L, median), respectively. The sample analysis results showed that the concentrations of PFHxS and PFOA in the intake water of M2 were 0.023 ± 0.076 μg/L (0.000 μg/L, median) and 0.011 ± 0.008 μg/L (0.008 μg/L, median), respectively, whereas those in the produced water were 0.054 ± 0.061 μg/L (0.031 μg/L, median) and 0.007 ± 0.008 μg/L (0.005 μg/L, median), respectively. During the monitoring period, PFOS was not detected in the intake or produced water of

either plant. The average concentrations of PFHxS in the produced water of both plants were higher than those in the intake water (0.052 μg/L for M1 and 0.054 μg/L for M2). The average concentrations of produced water in M1 and M2 increased by 125.3% and 43%, respectively. The PFHxS concentration in the treated water for both DWTPs was higher than that in raw water, indicating that there would be a source of PFHxS in water treatment processes for both M1 and M2. In contrast, the water treatment processes in M1 and M2 removed PFOA at 43% and 37.8%, respectively. The average concentration of PFOA in the treated water for both the DWTPs was 0.007 μg/L.

**Table 4.** Concentrations of PFHxS and PFOA in raw and produced water of M1 and M2.

| DWTP | | Parameters | Average | Median | S.D. | Min | Max |
|---|---|---|---|---|---|---|---|
| M1 | Inf. | PFHxS (μgL$^{-1}$) | 0.023 | 0.000 | 0.075 | 0.000 | 0.340 |
| | Eff. | | 0.052 | 0.028 | 0.052 | 0.007 | 0.231 |
| | Inf. | PFOA (μgL$^{-1}$) | 0.013 | 0.010 | 0.008 | 0.000 | 0.041 |
| | Eff. | | 0.007 | 0.005 | 0.009 | 0.000 | 0.034 |
| M2 | Inf. | PFHxS (μgL$^{-1}$) | 0.023 | 0.000 | 0.076 | 0.000 | 0.344 |
| | Eff. | | 0.054 | 0.031 | 0.061 | 0.006 | 0.267 |
| | Inf. | PFOA (μgL$^{-1}$) | 0.011 | 0.008 | 0.008 | 0.000 | 0.037 |
| | Eff. | | 0.007 | 0.005 | 0.008 | 0.000 | 0.034 |

The concentrations of PFHxS and PFOA were monitored by each unit process and operation used in M1 and M2 to identify the unit processes and operations that cause the increase in the concentration of PFHxS in treated water compared to that in raw water. As presented in Figure 2, both plants consisted of the same water treatment processes in the order of receiving well, pre-ozonation, coagulation/flocculation, settling basin, sand filtration, post-ozonation, activated carbon, and clear well. Evidently, PFCs were not easily removed by conventional coagulation/flocculation, sedimentation, and filtration because of their low concentration in the intake water and their hydrophilic characteristics. It has been reported that PFCs can be removed by activated carbon adsorption due to their physicochemical characteristics [27,28] but not by other treatment processes such as ozone oxidation, coagulation/flocculation, sedimentation, and filtration [29]. In particular, the strong C-F bonds and functional groups of -COOH and -SO$_3$H can make oxidation processes inefficient in removing them [10]. Figure 3 shows representative PFCs concentration profiles (data obtained in July) at each unit process and operation in M1 and M2. The influent PFHxS concentrations were 0.003 and 0.001 μg/L for M1 and M2, respectively. The concentrations of both compounds did not significantly change after post-ozonation (0.003–0.006 μg/L of PFHxS and 0.001–0.009 μg/L of PFOA in M1 and M2). However, contrary to the relatively high removal rate of PFCs by activated carbon [27,28], the concentrations of PFHxS in the GAC process increased by 13.2 times (from 0.006 to 0.079 μg/L) in M1 and 8.3 times (from 0.009 to 0.075 μg/L) in M2 (Figure 3). For M1, the PFHxS concentration in influent water increased by 26.3 times in the GAC process, from 0.003 μg/L to 0.079 μg/L. The same trend was observed in M2, with an increase of 75 times in the PFHxS concentration in the GAC process compared to the influent concentration (0.001 μg/L). The increase in PFHxS may be due to the higher desorption rate of PFHxS from the GAC than the adsorption rate. However, PFOA removal was not affected by the GAC process in either DWTP. The PFOA concentrations did not change much between the influent and effluent at 0.013 and 0.015 μg/L, respectively (Figure 3). Based on the observation of PFOA and PFHxS concentrations in the water treatment processes, it was assumed that the release of PFHxS from the GAC process (e.g., desorption of PFHxS) would be related to the GAC regeneration and replacement cycle in the process.

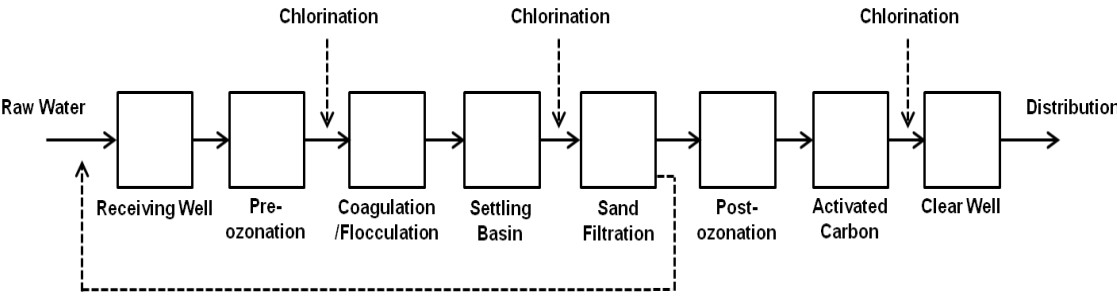

**Figure 2.** Schematic of water treatment processes of M1 and M2.

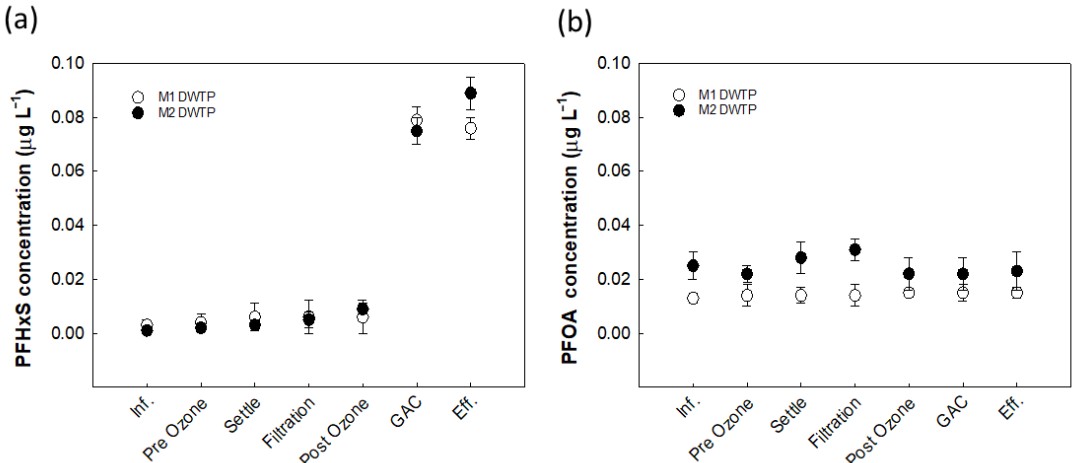

**Figure 3.** Concentrations of (**a**) PFHxS—perfluorohexyl sulfonate and (**b**) PFOA—perfluorooctanoic acid changed by the unit processes and operations of M1 and M2.

### 3.2. Effect of GAC Regeneration and Replacement on PFC Removal

In the GAC operation, the removal efficiency of PFCs seems to be closely correlated with the adsorption capacity of GAC. Generally, GAC must be replaced periodically to maintain sufficient adsorption capacity and remove target contaminants in DWTPs. GAC regeneration and replacement depend on the raw water quality (in terms of dissolved organic carbon (DOC), national organic matter (NOM), 2-methylisoborneol (MIB), and geosmin, etc.) and the adsorption capacity of GAC basins (or columns). It is known that used GAC may need to be replaced every one to two years.

The concentrations of PFHxS and PFOA in the GAC basins of M1 and M2 were analyzed to investigate the correlation between the removal efficiency of PFCs and the replacement cycle of GAC. The GAC systems were operated in parallel and treated with water supplied from the post-ozonation system. The maximum replacement cycle of GAC was three years for both M1 and M2, and the backwashing cycle lasted 5–7 days. Figure 4 shows the concentrations of PFHxS and PFOA in the water samples collected from the GAC basins of M1 (10 basins) and M2 (20 basins). The concentrations of PFHxS and PFOA in M1 were 0.086 ± 0.057 µg/L (0.052 µg/L, median) and 0.020 ± 0.010 µg/L (0.021 µg/L, median), respectively, while the concentrations of PFHxS and PFOA in M2 were 0.074 ± 0.049 µg/L (0.064 µg/L, median) and 0.009 ± 0.001 µg/L (0.009 µg/L, median), respectively. In the range of the GAC replacement cycle within 1–3 years, the PFHxS concentrations for each GAC basin for both M1 and M2 were significantly different up to 0.175 µg/L, which is 7.6 times higher than the raw intake water (i.e., 0.023 µg/L) for both M1 and M2. However, the PFOA concentrations were relatively consistent at 0.020 and 0.009 µg/L for M1 and M2, respectively, which were not significantly different from those of raw water.

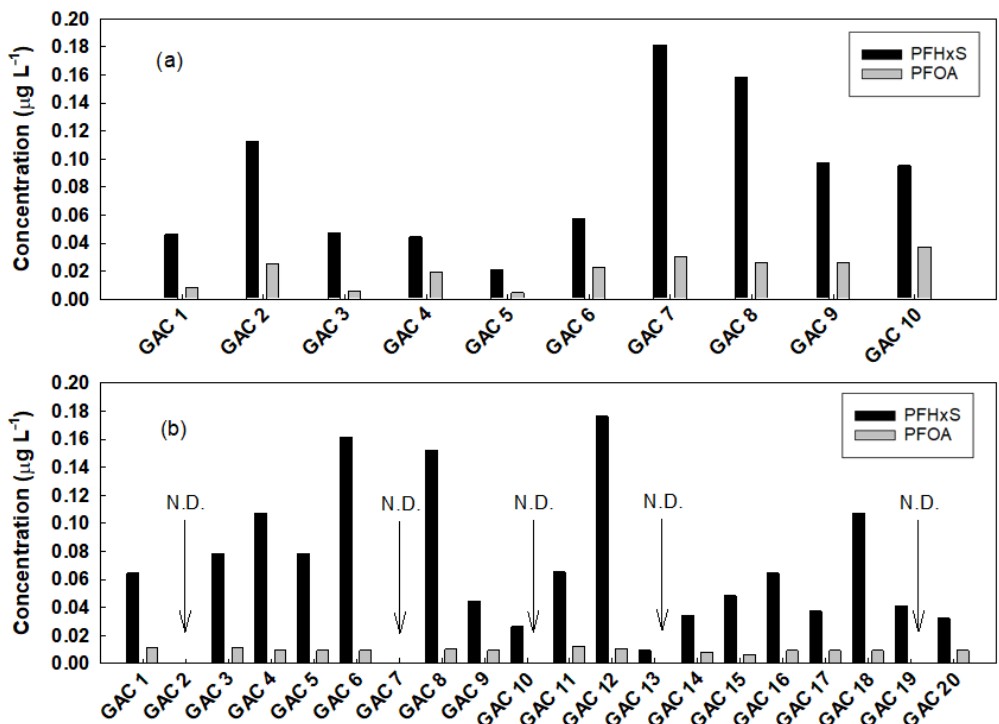

**Figure 4.** Concentrations of PFHxS and PFOA in the effluent of the GAC basins. (**a**) M1 and (**b**) M2. N.D: not detectable.

The PFHxS and PFOA concentrations were categorized based on the replacement cycle (i.e., <1 year, 1–2 years, and 2–3 years). As shown in Figure 5a, the concentration of PFHxS in the GAC process of M1 and M2 increased over time with a longer replacement cycle of GAC (0.015 ± 0.002 µg/L for <1 year, 0.051 ± 0.009 µg/L for 1–2 years, and 0.107 ± 0.008 µg/L for > 2 years). It was observed that GAC operation for less than one year ensured that PFHxS concentrations were lower than those of raw water, indicating that GAC sufficiently removed PFHxS (47.8% in this study). However, GAC columns over one year showed no effect on PFHxS removal. In contrast, the concentration of PFHxS increased to 0.051 and 0.107 µg/L compared to raw water (0.023 µg/L). In Figure 5b, although PFOA showed a trend similar to that of the GAC replacement cycle, its concentration change was negligible compared to PFHxS. The concentration of PFOA slightly increased as the replacement cycle duration increased (0.002 ± 0.005 µg/L for <1 year, 0.008 ± 0.003 µg/L for 1–2 years, and 0.009 ± 0.004 µg/L for >2 years). For both PFHxS and PFOA, the GAC operation for less than 6 months showed 100% removal of the compounds (GAC 2 and GAC 7 in Figure 4b). Thus, it was determined that a GAC replacement cycle of less than one year would improve the PFC removal efficiency, although the actual operation of the GAC regeneration and replacement cycle primarily depends on the receiving water characteristics of water utilities.

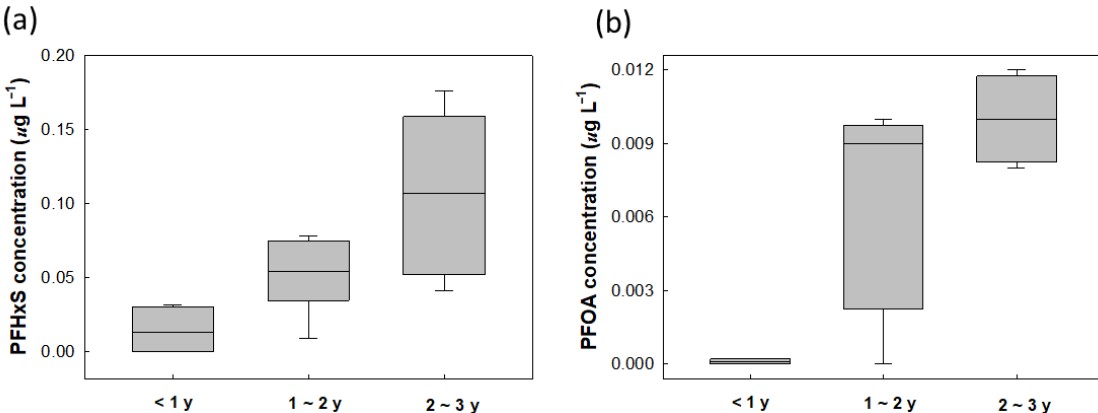

**Figure 5.** Concentration changes of (**a**) PFHxS and (**b**) PFOA in the effluent discharged from the GAC process by the GAC replacement cycle for M1 and M2.

### 3.3. Determination of a GAC Replacement Cycle for the Removal of PFCs

To determine the appropriate GAC regeneration and replacement cycle, a lab-scale coconut-shell-based GAC column test was conducted for 200 days using the same water after coagulation, flocculation, and sand filtration (Figure 1). The specific surface area and micropore volume of the coconut-shell-based GAC were 804.0 m²/g and 0.47 mL/g, respectively (Table 1). The other characteristics are listed in Table 1. Figure 6 shows the behavior of PFCs (the sum of PFHxS and PFOA in this study) throughout the experiment. PFHxS was removed at 63.7%, and breakthrough was observed at 108 days of the column test. The removal ratio of PFOA was 86.7%, and the breakthrough time was 161 days. An interesting observation was that the PFHxS breakthrough was advanced before the PFOA breakthrough in a mixture of PFHxS and PFOA of the test water. This finding agrees with other studies that showed high adsorption of PFOS to GAC. Similarly, the sulfonic functional group of PFHxS produces a greater electrostatic effect when compared to the carboxylic functional group found on PFOA [30]. The PFC concentration profiles indicated that PFHxS would be removed more quickly due to its sulfonic functional group that most strongly bonds with the surface of the GAC [30]. The total adsorption amount of PFCs was 1347.4 mg/ton-AC until the breakthrough, and the carbon usage rate (CUR) of GAC for PFHxS was 0.058 g/d, which was 57% greater than that of PFOA (0.037 g/d) (Table 3).

Removal efficiency using GAC may be affected by the compound's functional group(s) and PFCs' chain length. Zhang et al. studied the sorption of PFOS, PFOA, and perfluoro-heptanoic acid (PFHpA) on GAC [31]. The sorption rate was greatest for PFOS, followed by PFOA and then PFHpA, suggesting that the hydrophilic head group (sulfonate vs carboxylic) had an influence on sorption. Yu et al. compared feasibility of PAC and GAC for the removal of PFOS and PFOA [32] and found that the sorption mechanisms involve electrostatic, hydrophobic, and ion exchange interactions. Higher removal efficiencies for PFOS compared to PFOA were understood to be related to more hydrophobicity as PFOS has a longer perfluorinated chain. This is because the adsorbed PFOS molecules may prefer to adsorb other PFOS molecules over shorter chain PFOA to form hemi-micelles or micelles. However, from this study, it seems that higher removal of PFCs using GACs does not necessarily mean perpetual removal of the target contaminants. Although sulfonate groups resulted in a higher adsorption capacity compared to carboxylate groups [3], PFHxS showed the breakthrough compared to PFOA, indicating that the sulfonate-based PFCs (e.g., PFHxS) may be a limiting factor in GAC operations for PFCs removal.

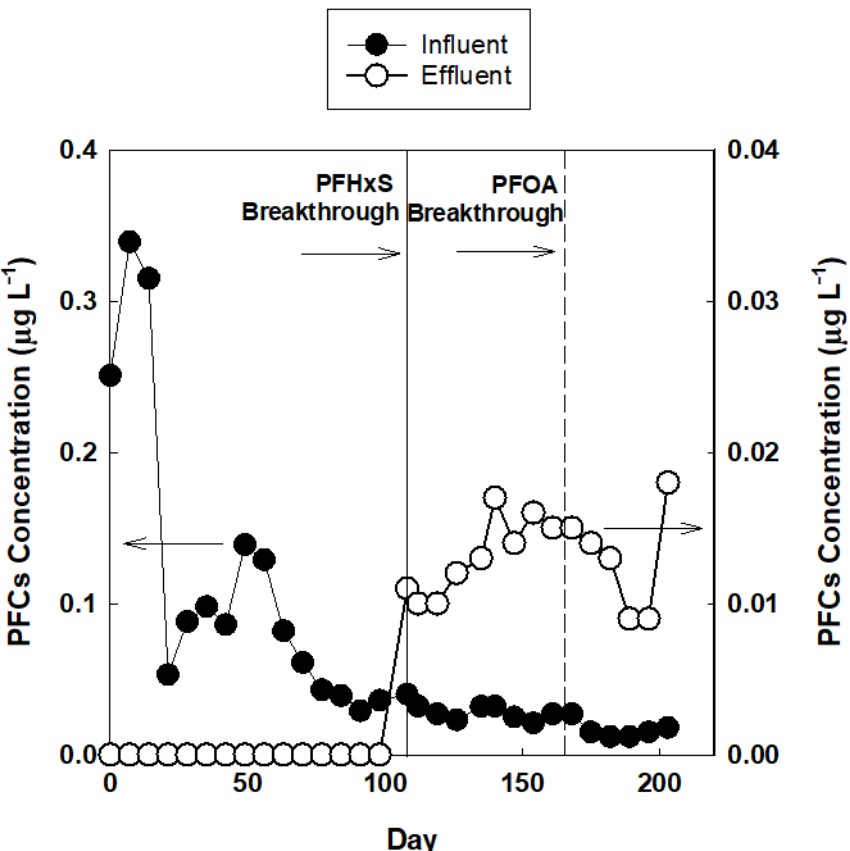

**Figure 6.** Concentration profile of perfluorinated compounds (PFCs) during the GAC column test.

## 4. Conclusions

GAC has been widely applied in DWTPs in the Republic of Korea due to its excellent micropollutant removal capability. Despite its removal efficiency, the concentration of PFCs increases through the GAC process. According to the analyzed data of PFHxS during this study, the average concentrations in the final product water of DWTPs M1 and M2 increased by 125.3% and 43.0%, respectively, compared to those of raw water. These phenomena can be explained by GAC desorption, particularly for PFHxS, and the significantly different concentrations in influent water might also affect the inverse results. Additionally, the replacement cycle of GAC had a significant effect on the removal efficiency of both PFHxS and PFOA. GAC with less than one year of the replacement cycle greatly improved the PFC removal efficiency, although the GAC regeneration and replacement cycle should be operated while considering the qualities and characteristics of other receiving waters. Considering the results obtained from the lab-scale experiments and the sample analysis of 30 GAC basins in two large-scale DWTPs, GAC replacement will be required within less than one year if the PFC concentrations in raw water are high.

**Author Contributions:** Investigation Y.-G.P., W.H.L. and K.K.; writing—original draft, Y.-G.P., W.H.L. and K.K.; writing—review and editing, Y.-G.P., W.H.L. and K.K.; All authors have read and agreed to the published version of the manuscript.

**Funding:** This study was supported by the Korea Agency for Infrastructure Technology Advancement (KAIA) grant funded by the Ministry of Land, Infrastructure, and Transport (Grant 21UGCP-B157942-02).

**Institutional Review Board Statement:** Not applicable.

**Informed Consent Statement:** Not applicable.

**Data Availability Statement:** Due to confidentiality agreements, supporting data can only be made available to bona fide researchers subject to a non-disclosure agreement.

**Conflicts of Interest:** The authors declare no conflict of interest.

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
