# Peer review of "Different Adsorption Behavior between Perfluorohexane Sulfonate (PFHxS) and Perfluorooctanoic Acid (PFOA) on Granular Activated Carbon in Full-Scale Drinking Water Treatment Plants"

_processes, doi:10.3390/pr9040571_

Round 1

Reviewer 1 Report

The manuscript presented for review is at an average scientific level. This manuscript should be completed to increase its scientific level. My suggestions are as follows:

  • Table 1 presents only a few parameters of the tested GAC coal. For example, other parameters of the porous structure should be completed: total pore volume, mesopore volume, pore size, etc. Other tests of the GAC material, showing its physicochemical properties, should also be completed.
  • Discuss the removal mechanism of perfluorohexane sulfonate (PFHxS) and perfluo-2 rooctanoic acid (PFOA) through granular activated carbon (GAC).
  • The results obtained in the work should be compared with the results presented in the literature. Discuss the obtained results in more detail.
  • No results are shown in Figure 5. This should be completed.
  • All references should be adjusted to the ‘’Processes’’ rules. This applies to the references in both the text and the References section.

Reviewer 2 Report

In this research work, PFHxS and PFOA contents have been analysed in two-large water treatment plants from Republic of Korea for eight months. Additionally, a lab-scale experiment has been performed to study and appropriate GAC replacement cycle.

Unfortunately, I do not recommend this work for its publication, although some studies are interesting and it is well written. Only in the case that this base manuscript is completed with more studies. In my opinion, the paper should be expanded and restructured on the basis of it.

General comments

  • The tittle of the manuscript is not totally consistent with the content of authors´ studies.
  • The conclusion of this work is not significant. It has no scientific interest, apart from the analysis of water samples from two DWTPs.
  • Authors just confirm that GAC should be replaced in order to improve the removal efficiency of PFHxS and PFOA compounds.
  • Is it operational, from an industrial point of view, the GAC replacement every 108 days?
  • They should have studied, for example, how to modify chemically the GAC or regenerate it in order to make it work for longer time.
  • Why did you select a coconut-shell-based GAC column? What is the type of GAC used in the DWTPs?
  • Note that information of Figure 5 has been lost. Table 1 is not with superscripts, etc…

Reviewer 3 Report

The aim of the paper is of high interest for water decontamination and the removal of persistent pollutants as PFCs, mostly, regarding the working time of GAC before the replacement.  However, Results and Discussion section is weak in merit of the data presentation, where standard deviation is higher respect mean value, this indicate that data set are spread out over a so large range of value, and taking in account of the reported data of concentration is not so accurate. On this basis, the depicted conclusions are not so strong.

Moreover, figures captions should be better explained to help readers for an immediate understanding of the plots, especially for Figure 2.

Other revisions refers to formal feature and omitted parts.

Line 178, please write sulphonil pedix -SO3H.

Line 146, please correct (from May to December).

In Figure 5, no plot appears on graphs.

In Figure 7, the concentration axis (on right and left side) lacks the symbol µ.

 I really appreciate the work, but it cannot be accepted in the present form. Please try to consider that sampling for a long interval of time can be affected of fluctuating composition of raw water and the evaluation on the whole population of data cannot be representative, organize subclass depending on a chosen parameter, for example the season, can be useful to get an reasonable set of data. From accurate value of the monitored concentration then conduct the discussion and fix conclusion.  

Round 2

Reviewer 1 Report

I have no comments. I accept the introduced corrections and supplemented the data.

Reviewer 2 Report

I rejected the paper and still think the same.

Reviewer 3 Report

After revisions the paper has been enhanced so I agree for publication.